# Spreading of Dangerous Skin-Lightening Products as a Result of Colourism: A Review

### Claudia C. A. Juliano [ID]

Dipartimento di Scienze Mediche, Chirurgiche e Sperimentali, University of Sassari, Viale San Pietro, 07100 Sassari, Italy; julianoc@uniss.it

**Abstract:** The use of bleaching products can have a medical or cosmetic purpose; in the latter case, skin whitening is most widespread in countries where darker skin tones prevail and can be driven by psychosocial, cultural and economic reasons. Skin-whitening products containing highly toxic active ingredients (in particular mercury derivatives, hydroquinone and corticosteroids) are easily found on the market; the use of these depigmenting agents can be followed by a variety of adverse effects, with very serious and sometimes fatal complications, and is currently an emerging health concern in many countries. This article concisely discusses the reasons for the current prevalence of skin lightening products and provides an overview of the skin lightening agents that pose a threat to human health. The review also reports market surveillance data on the circulation of banned skin lighteners in Europe, obtained through the Safety Gate system.

**Keywords:** skin lighteners; depigmenting agents; mercury; hydroquinone; topical corticosteroids; colourism; cosmetics; health risks; Safety Gate





## 1. Introduction

Melanins are polymorphous and multifunctional biopolymers that include eumelanin and pheomelanin; their synthesis in the skin, known as melanogenesis, occurs in melanocytes at the dermal-epidermal junction within intracellular organelles called melanosomes. Melanosomes are transferred via dendrites to the cytoplasm of surrounding keratinocytes, where they play a critical role in photoprotection. Melanogenesis is a complex process that comprises both enzymatic and chemical reactions and involves several enzymes: phenylalanine hydroxylase, tyrosinase (a glycosylated polyphenol oxidase containing copper), and tyrosine-related protein-1 (TRP-1) and TRP-2 (or dopamine tautomerase) [1]. Tyrosinase catalyses two distinct reactions in melanin biosynthesis: the hydroxylation of L-tyrosine to L-DOPA (L-3,4-dihydroxy-phenylalanine) and then the oxidation of L-DOPA to dopaquinone. Dopaquinone is highly reactive and, in the absence of thiol compounds, undergoes intramolecular cyclization, eventually leading to eumelanin; in the presence of thiols, such as cysteine or glutathione, it is converted to 5-S-cysteinyl DOPA or glutathionyl DOPA, eventually leading, through a complex series of reactions, to pheomelanin [1,2]. The pigments eumelanin (brown/black) and pheomelanin (red/yellow) both protect the skin from UV damage, although pheomelanin can also function as a powerful UV photosensitiser [3].

After UVB exposure, melanogenesis is induced by a variety of factors (cyclobutane pyrimidine dimers, alpha-melanocyte stimulating factor, stem cell factor, nitric oxide, adrenocorticotropic hormone or ACTH, endothelin-1) through different signaling pathways [1]; DNA repair also stimulates melanin production [4]. Melanin has a photoprotective effect by forming supranuclear caps in the cells of the human epidermis and thus reducing the formation of DNA photoproducts due to exposure to ultraviolet radiation; in addition, this pigment can chelate metal cations and exhibits antioxidant and radical-scavenging properties.

Skin lightening is defined as 'the practice of using chemicals or any other product with depigmenting potential in an attempt to lighten skin tone or improve the complexion; these goals are achieved by decreasing the concentration of melanin to achieve a reduction in the physiological pigmentation of the skin' [5]. Products used to achieve this purpose are known as depigmenting, skin-lightening, skin-bleaching, skin-brightening or skin-evening agents [6]. Several skin-lightening compounds have been developed and are now available for pharmaceutical and cosmetic purposes. Therapeutic indications for skin-lightening agents are generally aimed at the management of pigmentation disorders such as discolouration due to hormonal changes, melasma, age spots, senile/solar lentigo, post-inflammatory hyperpigmentation, and pigmented acne scars, as they reduce the hyperpigmentation of specific areas of the body and provide a more uniform skin colour. However, skin lightening is primarily a cosmetic procedure whose function is not only to lighten dark areas of the skin but also to achieve a generally lighter tone, particularly in countries where darker skin tones are prevalent and voluntary depigmentation meets the aesthetic criterion of a lighter skin [7].

Skin-lightening agents can operate through different mechanisms: the inhibition of tyrosinase transcription (e.g., tretinoin, retinol), the inhibition of tyrosinase (e.g., hydroquinone, azelaic acid, resveratrol), the acceleration of epidermal turnover (lactic acid, glycolic acid), the inhibition of melanosome transfer from melanocytes to surrounding keratinocytes, anti-inflammatory action and scavenging of free radicals, all of which have tyrosinase inhibition as their common goal [8,9]. Many effective depigmenting compounds available today can only be used as drugs (e.g., hydroquinone, retinoic acid) and should be restricted to use under dermatological supervision, while others are permitted in cosmetic products (e.g., alpha-hydroxy acids, arbutin, retinol, ascorbic acid). The development of new inhibitors of melanogenesis is of great importance in both the pharmaceutical and cosmetic industries. These inhibitors can originate from different sources, such as chemical synthesis and the screening of natural compounds and extracts [1]. In particular, the search for new whitening ingredients is driven by the demand for natural products, and traditional herbal derivatives are currently being investigated, as they are generally believed to be safer than synthetic compounds [5,9–11]. The different mechanisms of depigmenting agents legally authorised for medical and cosmetic purposes are summarised in Table 1.

**Table 1.** Mechanisms of action of legally authorised skin-lightening agents.

| Skin Lightening Agents | Mechanism of Action |
|---|---|
| Hydroquinone, Azelaic acid, Ellagic acid, Kojic acid, Mequinol, Arbutin, Flavonoids, Resveratrol, N-acetyl glucosamine | Inhibition of tyrosinase activity [9] |
| Alpha-hydroxy acids, salicylic acid | Acceleration of epidermal turnover [9] |
| Retinoids | Inhibition of tyrosinase transcription, epidermal melanin dispersion [8] |
| Vitamin C, Vitamin E | Antioxidant action, acceleration of epidermal turnover [9] |
| Niacinamide, Soy proteins, Linoleic acid | Inhibition of melanosome transfer [8,9] |

The present review aims to briefly discuss the reasons for the increasing popularity of skin-lightening products and then to make an overview of some skin-bleaching agents (mercury derivatives, hydroquinone, topical corticosteroids) that, although banned because of their toxicity, are still widely used in some countries, raising safety concerns. Finally, the review also reports recent data on the circulation on the European market of depigmenting cosmetics containing banned ingredients, obtained through the Safety Gate system.

## 2. Colourism and the Widespread Practices of Skin-Lightening

The belief that a fair complexion is a synonym for beauty and the widespread skin-lightening practices that result come from a complex interweaving of historical, cultural, social, psychological and economic factors.

Since ancient times, the canon of beauty has included a very fair complexion. During the 7th century in China, Empress We Zetian swallowed crushed pearls to obtain her fair complexion, a practice still popular among Chinese women. Greek women painted their faces with white lead (lead carbonate, very toxic) and Roman women adopted this practice using the same compound, which they called 'cerussa', as described by Pliny the Elder. Cleopatra (69–30 BC), queen of the Ptolemaic kingdom of Egypt, regularly bathed in acidic ass's milk [12]. In different historical periods, there have been many reasons behind skin-lightening practices; they are linked to colonialism in countries such as Africa or India and slavery in America [12].

Dangerous skin-whitening practices are not limited to the past, and the search for means to obtain a white complexion has continued unabated in the following centuries; striking examples are the ingestion of arsenic waffles in Victorian times, the use of radiotherapy at the beginning of the 20th century, or even, very recently, the oral or intravenous administration of glutathione recommended in some African countries to pregnant women to lighten the skin of babies in the womb [12]. Although Coco Chanel made the tanned complexion fashionable in the 1920s, there are still several psychosocial, cultural and economic reasons why people lighten their skin, because light complexion is still perceived as a positive value in many countries and cultures. This prejudice has its origin in so-called colourism; although colourism and racism are often interconnected, they are two different phenomena. Racism is prejudice against people of a certain ethnic group or "race", while colourism can be defined as "prejudice or discrimination against individuals with a dark skin tone, especially among people of the same ethnic or racial background", as defined by the Oxford English Dictionary.

Consequently, colourism places a higher value on light-skinned people than on darker-skinned ones. The historical reasons for colourism vary from country to country. In the United States, they are deeply rooted in the enslavement of Africans, while in South Africa, India and Latin America they are a consequence of European colonisation; on the other hand, in East Asia, light complexions were historically idealised and linked to wealth and desirability [7].

Currently, skin lightening is a practice that is mainly used in non-white communities around the world, including Africa, Asia, the Middle East and America; these countries now represent markets where demand for skin lightening products is strong. As reported by leading market research report provider Million Insights [13], the global skin lightening products market size is expected to reach $13.7 billion by 2025, and the CAGR (Compound Annual Growth Rate) is expected to grow at a rate of 7.4% from 2019 to 2025. The Asia-Pacific region is the largest market for this product type, accounting for 54.3% of global revenue in 2018.

Skin whitening is mainly practised by women, but in some geographical regions, it is also becoming popular among men and young adults [14]. At the moment, the largest number of studies on the spread of such products focuses on Africa. Skin whitening practices have been popular among African countries since the 1950s; the main motivation behind this phenomenon is the perception of greater economic and social privileges associated with fair skin.

It is estimated that about 75% of women in Nigeria, 60% in Senegal, 50% in Mali and 30% in Ghana regularly use bleaching products [15].

Compared to other African countries, South Africa has a lower rate of skin lighteners; after banning the use of hydroquinone in cosmetics and over-the-counter medicines in 1990, it became the first country in the world to ban skin lighteners, followed by Rwanda, Côte d'Ivoire, Tanzania, Kenya and Ghana [15].

Such practices are also very popular in Asian countries such as India, China, Japan, Korea and Arabia. In these cultures, having a fair complexion is considered an important element of female beauty, partly based on traditional values and partly due to the influences of Western colonial heritage [16]. In India, the words 'fair' and 'beautiful' are synonymous, and fairness is a distinctive feature.

The caste system is often blamed for creating a division based on skin colour in Indian society, with the light-skinned Brahmins occupying the apex of the caste pyramid and the darker-skinned Dalits ('untouchables') at the base [17]. Therefore, the sale of skin lighteners is still flourishing in India, and these cosmetics account for about half of all skincare products sold, with an approximate value of USD 500 million [18]. Asian skin is particularly prone to suffer from pigmentation disorders [8]; in a recent study by Porcheron et al. [19], Chinese women were asked to look at several images of Chinese faces before and after manipulation with graphic design software, and then to rate the age and attractiveness of the subjects depicted. The women who participated in the study perceived the faces as younger and more attractive after the manipulation of dark spots and dark circles than after the reduction of wrinkles and skin sagging [19].

Skin whitening is a global phenomenon; unfortunately, abuse of this practice is common. The availability and spread of formulations containing toxic, unapproved and/or illegal ingredients without a prescription is of particular concern. These substances, despite their apparent efficacy, are responsible for serious acute and chronic effects. The following paragraphs outline the skin-whitening compounds of greatest concern for human health. To find relevant sources of information, the author used well-known databases, including MEDLINE (PubMed), Science Direct and Google Scholar.

Keywords used to search for articles included colourism (or colourism), skin lightening products, skin whitening products, mercury derivatives, hydroquinone, topical corticosteroids and clobetasol. This research was carried out in the second half of 2021, and most of the articles consulted to write this review have been published in the decade 2010–2020. Government and official bodies' websites were also consulted to get information on legislative aspects.

### 3. Toxic Ingredients in Skin Lightening Cosmetics

*Mercury*

The main sources of mercury exposure in the past were industries (such as felt hat factories) and the composition and subsequent intake of mercury-based medicines. The use of mercury-containing skin-lightening products is currently a major cause of chronic mercury poisoning in some areas of the world [20]; despite their limited effectiveness, toxicity and the fact that they are banned in many countries, these products are still available and widely used worldwide.

They can be found in a range of creams, milks, lotions, gels, oils and soaps. Mercury occurs in three forms: elemental (or metallic) mercury ($Hg^0$), inorganic mercury compounds (existing in two oxidative states: mercurous, $Hg^+$, and mercuric, $Hg^{++}$) and organic mercury compounds. Inorganic mercury (mercurous chloride or calomel, $Hg_2Cl_2$; ammoniated mercury, $HgNH_2Cl$; mercuric iodide, $HgI_2$; mercurous oxide, $Hg_2O$; mercuric chloride, $HgCl_2$) is the one commonly found in skin-lightening preparations [21], although methylmercury has recently been reported to be present in a skin-lightening product in the United States [22]. Mercury compounds suppress melanin production in skin melanocytes. Tyrosinase, a type III copper protein, plays a key role in the synthesis of this pigment because, as already mentioned, it catalyses the first two reactions of melanin production: the hydroxylation of the amino acid L-tyrosinase to DOPA and then the conversion of DOPA to dopaquinone. Mercurous and mercuric ions inhibit melanin synthesis by competing with copper in tyrosinase; in particular, they bind to the His residue of the catalytic centre of the enzyme. This inhibition is irreversible [23].

Humans are exposed to mercury through dermal absorption, inhalation of elemental mercury vapours (especially in cases of long-term exposure) and ingestion. Inorganic

mercury salts contained in creams and other cosmetics are easily absorbed through the skin; they penetrate the epidermis and are also absorbed through sweat glands, sebaceous glands and hair follicles [24], through which Hg eventually accumulates in the hair of the scalp [25]. Abbas et al. [25] studied Hg exposure in students using cosmetics to lighten their skin and found that Hg concentrations in the hair had a statistically significant correlation with the Hg concentration in cosmetics. Dermal absorption depends on factors such as mercury concentration, frequency of product application, skin integrity, lipophilicity of the vehicle in the cosmetic product and the hydration of the stratum corneum; ingestion of mercury can occur after topical application around the mouth and hand-to-mouth contact [26].

Inorganic mercury compounds accumulate in the body's organs, with the highest concentration found in the kidney, near the proximal tubule; urinary excretion is the main route of elimination [24]. Since inorganic mercury compounds are not lipophilic, they do not readily pass through the blood-placenta or the blood-brain barriers [24]; however, prolonged dermal exposure can lead to accumulation in the central nervous system and neurotoxicity [27]. Inorganic mercury salts are mainly excreted in urine and faeces; high levels of urinary mercury are frequently associated with the use of mercury-containing skin lighteners.

According to current information on exposure to cosmetics for commonly used products (1.22 g for a face cream and 4.42 g for a body lotion; [28]), a consumer using a cosmetic containing 10,000 ppm mercury could absorb up to 450 µg in a single application [29]. Increased and often elevated concentrations of mercury in urine have been found in people who used skin-lightening products and also in family members who were not using them [27,30,31]. The inhalation of elemental mercury vapours depends on the transformation of inorganic mercury compounds into elemental mercury, which can occur in the presence of low pH and UV radiation [32,33]. The following reaction shows the dissociation of calomel into mercuric chloride and elemental mercury, which evaporates into the air:

$$Hg_2Cl_2 \rightarrow HgCl_2 + Hg^0$$

In vapour form, elemental mercury is lipid-soluble and highly diffusible through cell membranes; it is easily absorbed in the lungs, but also through the nose by the olfactory pathway [24], enters the bloodstream and rapidly spreads throughout the body, crossing the blood-brain barrier and accumulating in the central nervous system. Elemental mercury is also oxidised to mercuric form in tissue cells [24]. This reaction shows how the prolonged use of creams containing mercury salts can spread elemental mercury within the household via contaminated items (clothes, towels) or surfaces. Copan et al. [27] reported several cases of widespread household contamination in Mexico and California. Mercury vapour levels detected in bedrooms ranged from undetectable to 8 µg/m$^3$; these cosmetics may therefore contaminate people living in the same house, notably children and the elderly.

It is worth mentioning that, according to health-based risk levels for elemental mercury in the air set by regulatory agencies (e.g., ATSDR, Agency for Toxic Substances and Disease Register), 1 µg/m$^3$ is the indoor air level below which staying in an environment is considered safe and no remediation is required, while 10 µg/m$^3$ is the level above which evacuation of residents is recommended; at levels ranging from 1 to 10 µg/m$^3$ evacuation is not required but remediation is recommended [27].

It is worth mentioning that, according to health-based risk levels for elemental mercury in air set by regulatory agencies (e.g., ATSDR, Agency for Toxic Substances and Disease Register), 1 µg/m$^3$ is the indoor air level below which staying in an environment is considered safe and no remediation is required, while 10 µg/m$^3$ is the level above which evacuation of residents is recommended; at levels ranging from 1 to 10 µg/m$^3$, evacuation is not required but remediation is recommended [27].

The toxic effects of topically applied mercury-containing products have been documented since the beginning of the 20th century. Acute or chronic exposure can result in dermal, gastrointestinal, neurological and renal toxicity. Organic and metallic mercury, which is more lipophilic, is more typically associated with neurological damage,

while inorganic mercury more often causes kidney damage. Dermatological effects include allergic contact dermatitis, redness, erythroderma, nail discolouration, purpura and paradoxical hyperpigmentation [34].

Gastrointestinal symptoms include a metallic taste in the mouth, gingivostomatitis, nausea and hypersalivation. Neuropsychiatric signs and symptoms are the most significant indicators of mercury poisoning due to the use of skin brighteners; the most frequent are tremor, muscle weakness, peripheral neuropathy, depression, psychosis, anxiety, dizziness, headache and vision loss [35]. Regarding renal damage, acute exposure to mercury usually causes tubular injury (acute tubular necrosis), while chronic exposure is more frequently responsible for glomerular injury [34]. A common clinical sign in exposed infants and children is hypertension [27]. Since mercury crosses the placenta, the use of skin lighteners containing mercury during pregnancy may lead to pre- and post-natal intoxication, leading to serious harmful effects on babies [35,36].

Several studies have evaluated mercury levels in popular skin-whitening products, demonstrating that these cosmetics are an important exposure pathway to Hg and a threat to human health. For example, Peregrino et al. [37] analysed 16 locally produced Mexican skin-whitening creams using cold vapour atomic absorption spectrometry (CV-AAS). Mercury was detected in six of the 16 samples, in amounts ranging from 878 to 36,000 ppm. In a 2014 investigation, Hamann et al. [29] investigated 549 US skin-lightening cosmetics, both online and in shops, using X-ray fluorescence spectrometry. Of the products tested, 6% contained mercury above 1000 ppm and 45% contained mercury above 10,000 ppm. Copan et al. [27] reported that mercury levels of artisanal lightening creams from Mexico containing calomel ranged from 28,000 to 210,000 ppm. More recently, Ricketts et al. [22] used X-ray fluorescence and cold vapour atomic absorption spectroscopy to assess mercury concentrations in 60 popular lightening cosmetics on the Jamaican market. Mercury concentrations ranged from 0.05 ppm to 17,547 ppm; six contained mercury above 1 ppm, the maximum allowable limit set by the US Food and Drug Administration (FDA), and three were characterised by a particularly high mercury presence (17,547, 465.73 and 422.04 ppm respectively). Creams contained more mercury than soaps and lotions. On the other hand, a survey of 62 lightening creams and soaps on the Ghanaian market revealed that mercury levels in all samples tested ranged from <0.001 to 0.327 $\pm$ 0.062 μg/g and were therefore below FDA limits [38].

In conclusion, despite several restrictive regulations, skin-whitening preparations containing mercury continue to be available on the global market, and without geographical limitations; the public continues to have access to these products even in countries with stronger market surveillance, as they can be purchased online, in ethnic markets, from abroad and even from flea markets. Furthermore, the results of these investigations indicate that the mercury content of skin-whitening cosmetics is sometimes extremely high and poses a serious health threat, putting consumers and their families at risk of poisoning.

Mercury is sometimes mentioned on the packaging and its concentration is occasionally indicated, but in many cases, it is not even included among the ingredients, so consumers are not necessarily well informed by what is on the label. For these reasons, doctors should be aware that mercury-containing skin-lightening cosmetics, although illegal, are still available on the market, and that symptoms (dermatitis, weakness, myalgias, change of taste, paresthesias) of unclear cause could be related to mercury poisoning, even when patients claim not to use them [29].

## 4. Hydroquinone

Hydroquinone (1,4-dihydroxybenzene; Figure 1) is a water-soluble phenolic compound in the form of colourless or white crystals; it is a ubiquitous molecule and is found in tea, coffee, beer, berries, propolis and some mushrooms.

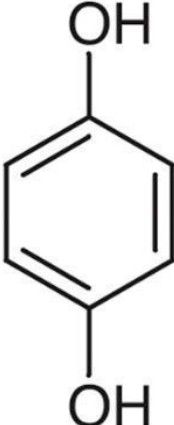

**Figure 1.** Hydroquinone structure.

A natural derivative of hydroquinone is α-arbutin, a glycoside in whose structure the hydroquinone is bound to a D-glucose molecule [38] (Figure 2).

**Figure 2.** Arbutin structure.

Arbutin exists in aqueous solutions in α, β, or γ-anomeric form, with β-anomer being the dominant form. Arbutin had skin-lightening properties due to the inactivation of tyrosinase, and possesses antioxidant properties that can contribute to its depigmenting action [39]. Although there is the possibility that a small amount of hydroquinone can be produced by cutaneous microorganisms or UV radiations when arbutin is applied to the skin, this molecule has intrinsic depigmenting properties, that are not dependent on the release of hydroquinone [39].

There are many industrial uses for hydroquinone: it is used in photographic developers as a polymer inhibitor, as an antioxidant for fats and oils, and as an intermediate for rubber processing chemicals. It also serves as a depigmenting agent in concentrations ranging from 2% to 10% [40].

Interest in hydroquinone as a skin-lightening agent arose in the 1960s following the accidental discovery of its skin-whitening effect on black American workers who were exposed to it on a daily basis in the rubber industry [8].

Hydroquinone's use for depigmenting purposes became widespread in the following decades and is currently utilised in managing various hyperpigmentation disorders such as melasma, chloasma, freckles, age spots and post-inflammatory hyperpigmentation caused by acne or trauma [34]. Hydroquinone is present in numerous prescription-only preparations; for example, in the Netherlands, pharmaceutical wholesalers supplied pharmacies with 90 kg of hydroquinone and it is estimated that 5000–10,000 formulations of 2% hydroquinone and 10,000–15,000 formulations of 5% hydroquinone were prescribed in a single year [40]. Although hydroquinone has been the dermatological gold standard for skin lightening for over 50 years, in recent times regulatory agencies in Japan, Europe and

the United States have questioned its safety [41]. Considering the risks to human health associated with its application, hydroquinone has been banned from cosmetics in Europe since 2001 [40] and its use in skin-lightening cosmetic formulations is currently illegal (see the "Regulations" section for more details). Cosmetics containing hydroquinone have also been banned in several other countries, such as the United Kingdom, Australia and Japan.

Several researchers have focused their attention on the mechanisms of depigmentation produced by hydroquinone. It has been shown that hydroquinone inhibits the activity of the tyrosinase enzyme by blocking the oxidation of tyrosine to DOPA and the subsequent oxidation of DOPA, causing the cessation of melanin synthesis; hydroquinone treatment decreases the number of melanised melanosomes and leads to the formation of abnormally melanised melanosomes, which eventually die [42]. Recently, Aspengren et al. [43] demonstrated that tyrosinase is not the only cellular target of hydroquinone, since this compound severely affects certain cytoskeletal structures (microtubules, actin filaments) in cultured melanophores of Xenopus laevis in a dose-dependent manner. Depigmentation by hydroquinone is reversible; after its suspension, melanin synthesis resumes [43].

For depigmentation purposes, hydroquinone is generally applied once a day in the form of a 2–10% cream; treatment is normally limited to the face, décolleté and back of the hands, but in some cases the compound is used on other parts of the body, reaching up to 5% of the body surface [40]. Studies on dermal absorption, carried out both in in vitro models and in in vivo experiments on humans, have shown that hydroquinone easily and rapidly penetrates human skin, although percutaneous absorption may be affected by the vehicle [44,45]; following absorption, it is found in the bloodstream and occurs partly as free hydroquinone, partly reversibly bound to proteins, and finally irreversibly bound to proteins [40]. It is excreted via the urine, but in a slower manner than in the absorption phase. Hydroquinone is metabolised in the liver as glucuronide and sulphate and oxidised to p-benzoquinone, especially in the bone marrow. It is known that p-benzoquinone causes specific mutations and is an initiator of carcinogenesis [46].

The use of skin-lightening products containing hydroquinone can cause short- and medium-term effects, both acute and chronic. The most common acute complication is irritant contact dermatitis [34]. Chronic side effects due to hydroquinone exposure are of more concern and include ochronosis, nail discolouration, conjunctival melanosis and corneal degeneration [47]; however, ocular complications only occur in individuals exposed to atmospheric hydroquinone and have never been reported after topical application of the compound. The most common complication induced by prolonged exposure to hydroquinone is exogenous ochronosis, a localised hyperpigmentation of the skin with asymptomatic blue-black and grey-brown macules with no systemic manifestations, histologically characterised by banana-shaped ochre deposits and irregularly shaped collagen bundles in the dermis [48,49]. Although the aetiology of this hyperpigmentation is unknown, it has been suggested that hydroquinone may inhibit homogentisic acid oxidase in the dermis, resulting in local accumulation of homogentisic acid which then polymerises to form ochronotic pigment [47]. A unique complication of chronic hydroquinone use is trimethylaminuria or 'fishy smell syndrome', characterised by an unpleasant rotten fishy odour due to the excretion of trimethylamine in sweat, urine, saliva, vaginal secretions, and breath [34]. Hydroquinone probably induces this effect by reducing the enzymatic oxidation of trimethylamine to trimethylamine N-oxide [50].

The carcinogenicity and genotoxicity of hydroquinone are so well established in rodents that an association between long-term use of this compound as a skin brightener and squamous cell carcinoma in humans has been suggested [51–53]. However, regulatory authorities have concluded that there is insufficient data to classify hydroquinone as a carcinogen. For example, the American Conference of Governmental Industrial Hygienists classified hydroquinone as 'A3', which means that it is recognised as carcinogenic in animals but the relevance for humans is unknown [47]. Similarly, the International Agency for Research on Cancer (IARC) has classified hydroquinone in Group 3 (not classifiable as a human carcinogen); hydroquinone is not classifiable as a human carcinogen, based on

limited evidence in experimental animals and inadequate evidence in humans. In the EU, hydroquinone is classified as Carc. Cat 2 H351 (suspected of causing cancer) and Muta. Cat 2 (suspected of causing genetic defects) according to Regulation (EC) No 1272/2008 (CLP Regulation), Annex VI1 [54].

Concerning the safety of arbutin, it is interesting to note that the Scientific Committee on Consumer Safety (SCCS), in its opinion on α-arbutin [55], stated " . . . the use of arbutin in skin whitening products is a complex situation for which the level of application and local availability of hydroquinone cannot be generalised. The factors causing the release of hydroquinone on the skin surface depend on the activity of bacteria and skin enzymes (glycosidases) in each individual and there is insufficient information on their quantitative variation. In addition, the dermal penetration of the active substance and its stability in formulations may vary. Therefore, a case-by-case evaluation is required' [55]. SCCS concluded that " . . . the use of α-arbutin is safe for consumers in cosmetic products in a concentration of up to 2% in face creams and up to 0.5% in body lotions" [55].

## 5. Corticosteroids

Topical corticosteroids are among the most widely prescribed drugs in clinical dermatology; they are used in the management of a wide range of medical conditions, due to their anti-inflammatory, antimitotic and immunomodulatory actions [56]. Abuse of topical corticosteroids, with the aim of achieving lighter skin, is a widespread practice in many countries, such as India and sub-Saharan African states, especially among women [6,50,57,58]. The misuse of these products is facilitated by their availability as cheap over-the-counter (OTC) drugs [58].

They are used as skin brighteners due to their potent depigmenting action and anti-inflammatory effects, which are useful in reducing the irritative potential of other whitening products that might be applied in combination [57]; clobetasol propionate, betamethasone dipropionate and fluocinonide (fluocinolone acetonide) are the most commonly used agents [12]. However, the mechanisms of corticosteroid-induced skin depigmentation are not yet fully understood. Several explanations have been proposed, including a direct cytotoxic effect, vasoconstriction, mechanical effects of oedema or an alteration in the regulation of melanogenesis [59]. Their lightening effect is initially thought to be mediated by local vasoconstriction, which gives the impression of an immediate reduction in skin pigmentation [20]; finally, corticosteroids lighten the skin by inhibiting pro-opiomelanocortin (POMC), a protein synthesised in the anterior pituitary that produces, by proteolytic cleavage, several biologically active peptides, including α-melanocyte-stimulating hormone (α-MSH), which regulates melanocyte function [34,50].

In order to exert their lightening action, corticosteroids are applied at high concentrations and over a large area of the body for prolonged periods (from a few months to a few years); long-term use and the conditions of application favour their dermal absorption [50]. The cosmetic use of corticosteroids is associated with a wide range of side effects, both dermatological and systemic; these adverse effects are directly connected to the potency of the compounds, since steroids used as lighteners are generally classified as potent or very potent [57]. Skin complications include acne vulgaris, allergic contact dermatitis, skin atrophy, hypertrichosis and telangiectasias. In addition, topical corticosteroids predispose to skin infections, such as dermatophytosis, folliculitis, erysipelas, scabies and viral warts [34]. Systemic adverse effects due to chronic corticosteroid use include Cushing's syndrome, diabetes mellitus, immunosuppression, hypertension, and suppression of the hypothalamic-pituitary-adrenal axis with adrenal suppression, the latter being the most alarming complication, as it can lead to death [34].

This side-effect should not come as a surprise, since literature has shown that the maximum monthly amount of clobetasol propionate cream (0.05%) applied to the skin for whitening purposes could reach 480 g [60]; more than 50 g of this cream per week can induce a profound suppression of the hypothalamic-pituitary-adrenal (HPA) axis [61]. Finally, chronic daily application of steroids can lead to the so-called "steroid dependency syn-

drome", characterised by an intense burning and potentially permanent erythema ("*Homme rouge*") following the discontinuation of steroid treatment [34]; this syndrome occurs more frequently in male users and can only be alleviated by reapplication of the drug [50].

The mechanisms of action and the most common undesirable and toxic effects of the banned lightening agents examined are summarised in Table 2.

**Table 2.** Mechanisms of action and toxic effects of the banned depigmenting agents examined.

| Skin-Lightening Agent | Mechanism of Action | Toxic Effects |
|---|---|---|
| Mercury derivatives | Inhibition of melanin synthesis (competition with copper in tyrosinase's activity site) | Dermatological, gastrointestinal, neurological and renal effects |
| Hydroquinone | Inhibition of tyrosinase activity | Irritant contact dermatitis, ochronosis; suspected of causing cancer |
| Corticosteroids | Interference with the production of $\alpha$-MSH | Dermatological (acne, contact dermatitis, skin atrophy) and systemic (Cushing's syndrome, immunosuppression, hypertension, diabetes mellitus, hypothalamic-pituitary-adrenal axis suppression) effects |

## 6. Regulations

The Minamata Convention on Mercury, adopted in 2013 [62], is a global treaty that aims to protect human health and the environment from the adverse effects of mercury; one of its goals is to phase out mercury in a large number of popular products. By the end of 2020, parties to the Convention will have banned the production, import and export of soaps and skin-lightening creams containing more than 1 ppm (1 mg/kg) of mercury, while no limit value has been set for eye cosmetics in which mercury derivatives are used as preservatives and for which there are currently no effective and safe alternatives available.

To protect European citizens, EU Regulation 1223/2009 [63] banned the use of hydroquinone, mercury and topical steroids in skin-whitening cosmetics; in particular, corticosteroids and mercury and its compounds (except phenylmercury salts, listed in Annex V of Regulation 1223/2009 and which are allowed as antimicrobial preservatives in eye products) are present in Annex II, the list of substances prohibited in cosmetic products. Concerning hydroquinone, it was initially listed in Annex III of EU Cosmetic Regulation 1223/2009 (the list of substances that cosmetic products must not contain except in specific cases) and was only allowed as an oxidising dye for professional hair dyes (maximum concentration 0.3%). Subsequently, with Regulation 344/2013 [64], the use of hydroquinone was further restricted, and it was included in Annex II (substances whose use is prohibited in cosmetics), except for entry 14 of Annex III, where hydroquinone is no longer allowed in hair dyes but only in artificial nail systems for professional use (maximum concentration 0.02%). The intention of the community legislator therefore seems to be the progressive exclusion of hydroquinone from cosmetic products, thus narrowing its use to professional use products that do not involve direct contact with the epidermis.

In the United States, the 1974 FDA regulation [65] states that any cosmetic product that contains more than unavoidable trace amounts of mercury (defined as no more than 1 part per million [1 ppm = 0.0001 per cent] calculated as a metal) is in violation of the Federal Food, Drug and Cosmetic (FD&C) Act and is subject to legal action. The only exceptions are cosmetics intended for use only in the eye area, which may not contain more than 65 ppm (0.0065%) of mercury, calculated as a metal, as a preservative, when no effective and safe alternatives are available. As for hydroquinone, in 2014 the U.S. Cosmetic Ingredients Review (CIR) Expert Panel [66] stated that this ingredient was safe for use in nail adhesives, but should not be allowed in other leave-on cosmetic products. Recently, based on the CARES Act (2020; [67]), all OTC products containing hydroquinone have been withdrawn from the US market, and products containing this ingredient must be approved by the FDA via the new drug that process.

### 7. Notifications on Safety Gate

The European Commission's EU Rapid Alert System for Dangerous Non-Food Products (formerly known as RAPEX) [68] enables the rapid exchange of information on non-food products on the European market that pose a risk to the health and safety of consumers. This system has a dedicated public website, the "Safety Gate", which provides access to weekly updates of alerts submitted by national authorities participating in the system. We decided to run a search on this website on skin lighteners available between 2011 and 2020 on the European market and identified them as not complying with Regulation 1223/2009 due to the presence of non-permitted ingredients. The search criteria adopted were: the name of the active substance (mercury, hydroquinone, clobetasol, betamethasone, fluocinolone) entered in the free text search box, the product category ('Cosmetics'), the hazard type ('Chemical'), the reporting type ('All') and the product user ('All'). According to the results of this survey, in the decade 2011–2020, 234 cosmetic products intended for skin-lightening were reported as presenting a chemical risk, due to the presence of mercury, hydroquinone or corticosteroids (Table 3). In some cases, the banned substances were listed on the packaging, but very often the label incorrectly listed the ingredients and omitted the illegal ones. It is important to keep in mind that, although most of the notified products were available in physical shops, some were also sold online (especially via the eBay platform).

**Table 3.** Skin-lighteners not complying with Regulation 1223/2009 notified in the period 2011–2020 on the Safety Gate website (* Numbers in brackets refer to betamethasone notifications).

| Year | Mercury | Hydroquinone | Corticosteroids * |
|------|---------|--------------|-------------------|
| 2020 | 9 | 8 | 6 |
| 2019 | 15 | 6 | 2 (1) |
| 2018 | 0 | 8 | 1 |
| 2017 | 0 | 8 | 0 |
| 2016 | 0 | 12 | 2 (1) |
| 2015 | 0 | 19 | 0 |
| 2014 | 2 | 12 | 8 (1) |
| 2013 | 30 | 3 | 2 |
| 2012 | 4 | 22 | 7 (1) |
| 2011 | 1 | 42 | 5 |

The number of mercury-containing cosmetics notified on the European Safety Gate in the decade 2011–2020 was 61; almost all of these products (54) were creams, and only two were soaps, while the type of formulation was not declared for five of them. The range of mercury concentrations (declared value on the packaging or measured amount) was 0.4 mg/kg–38,800 mg/kg (3.88% by weight); 34 products contained more than 0.5% (5 g/kg) by weight of mercury and 20 contained more than 1% (10 g/kg). In terms of their geographic origin, most of these cosmetics were produced in Pakistan (52.5%) and the People's Republic of China (23.0%), while 11.5% were of unknown provenance.

On the Safety Gate portal, 140 notifications concerning skin lighteners containing hydroquinone were published in the years 2011–2020 (five notifications of cosmetics containing hydroquinone concerning glue for false eyelashes and glue for false nails were not included here). Of the notified products, 68 were creams, 46 were body milks or lotions, 11 were oils, five were soaps, two were gel creams, and one was a gel; the type of the remaining seven products was not specified.

The concentrations of hydroquinone detected or declared on the label ranged from 0.01% to 9.6% by weight; 52.9% of the notified products contained at least 1% hydroquinone, and 8.6% contained at least 5%. Concerning their countries of origin, 32.9% were from

Côte d'Ivoire, 10% were from the United States, 7.1% were from France, 6.4% were from the Democratic Republic of Congo, 3.6% were from the United Kingdom, and 7.1% were from other countries, mainly in Africa; the origin of the remaining 27.9% was not identified.

As a final observation, 33 notifications concerning cosmetics containing corticosteroids were submitted; in most cases, the steroid was clobetasol, which was present in 29 preparations, while four contained betamethasone (in concentrations ranging from 0.0193% to 0.054%), and no product contained fluocinolone. Clobetasol was always present as propionate, and its concentrations, either declared or detected, ranged from 0.0094% to 0.06% by weight; 11 cosmetics contained 0.05% clobetasol propionate, while no indication was given on the labels of 8 of them. The majority of the products (69.0%) were creams; the most frequent country of origin was Côte d'Ivoire (37.9%), followed by Italy (20.7%); in 13.8% of the cases, there was no indication of geographical origin.

The analysis of the Safety Gate notifications confirms that, despite the strict and well-enforced regulation of cosmetics in Europe, a significant number of skin-lightening products containing non-permitted ingredients are available on the market.

## 8. Conclusions

The circulation on global markets of lightening cosmetics containing illegal and toxic ingredients is now a growing public health concern; products containing banned ingredients are readily available in many countries and sold both in traditional shops and online, despite all of the strict regulations currently in force. Addressing this problem requires a multi-faceted approach, as it encompasses several aspects. First, existing national regulations should be more strictly enforced, and more should be done at an international level to reduce the circulation of illegal skin whitening products, especially with regard to online sales.

Secondly, further studies are needed on the toxicity of skin-lightening ingredients of major concern for human health, since final users often apply self-made mixtures of different products with sometimes unpredictable toxic effects. Scientists should also expand their focus to the long-term health effects of undue skin-lightening practices, such as cancer; it cannot be excluded that an excessive inhibition of melanogenesis can result in an increased susceptibility to skin cancer in darker populations living in tropical countries due to UV radiation.

Moreover, in this era of rapid population migration and the consequent spread and globalisation of customs and traditions, every healthcare professional should be informed about skin-whitening practices and their potential negative health effects so that they can inform and protect their clients and patients and identify clinical conditions related to the application of toxic skin-whiteners. Awareness and knowledge regarding the issues related to the use of dangerous skin lighteners is essential to determine the cause of the disease and to initiate appropriate treatment in a timely manner.

Lastly, the public's understanding of this problem should be promoted among populations and communities that routinely practice skin whitening; for this purpose, an adequate information campaign and public education should be developed to raise awareness of the risks of using inappropriate products. This should be done by keeping in mind the cultural and socioeconomic reasons why people lighten their skin and therefore offering solutions that make proper cultural sense.

**Funding:** This research was founded by FAR2019Juliano found (University of Sassari, Italy).

**Institutional Review Board Statement:** Not applicable.

**Informed Consent Statement:** Not applicable.

**Data Availability Statement:** Not applicable.

**Conflicts of Interest:** The author declares that they have no conflict of interest.

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
