# Peer review of "Spreading of Dangerous Skin-Lightening Products as a Result of Colourism: A Review"

_applsci, doi:10.3390/app12063177_

Round 1

Reviewer 1 Report

Journal Name: Applied Science

Title: Diffusion of dangerous skin lightening products as a result of colorism: a review

In the current review article, the effect of the skin lightening products ingredients (mercury and hydroquinone) affect human health.  

Article accepted in present form or with minor corrections.

  1. The author should trim the abstract
  2. Authors need to refer few earlier articles of Applied Science and explain why the article is suitable for Applied Science Journal
  3. The authors need to further improve the literature review.
  4. Why author has selected mercury not other heavy metal ions like arsenic, tin, and lead.
  5. Why author have selected hydroquinone not other similar chemicals molecules like resorcinol and catechol.
  6. The conclusion can be further improved
  7. What should be the minimum concentrations of the Hg and hydroquinone present in the lightning products?
  8. What are all the available methods to detect Hg and hydroquinone in commercial samples?
  9. Please check the grammatical and syntax error

Author Response

Author’s Reply to the Review Report (Reviewer 1)

First of all, thank you for taking the time to assess my manuscript.

  1. The author should trim the abstract.” According to your suggestion, the abstract has been shortened.

  1. “Authors need to refer few earlier articles of Applied Science and explain why the article is suitable for Applied Science Journal”. The growing popularity of skin-lightening cosmetics and their possible impact on human health have led in recent times a flourishing of scientific studies in this field. The Author felt that this review was appropriated to Applied Sciences, Section Applied Biosciences and Bioengineering, because its subject matter is topical and societally relevant.

  1. “The authors need to further improve the literature review”. The bibliographical section of the original manuscript was updated to January 2022; however, a couple of new references have been added.

  1. “Why author has selected mercury not other heavy metal ions like arsenic, tin, and lead.” and “Why author have selected hydroquinone and not other similar chemicals molecules like resorcinol and catechol”. The purpose of the review was to describe the toxic substances more present in skin lighteners popular in countries where the skin bleaching has not only a purely cosmetic meaning, but also has profound cultural and socio-economic implications. In skin-lightening formulations circulating in these countries, the banned ingredients traditionally employed are mercury derivatives, hydroquinone, and corticosteroids; no other heavy metals or phenols have been detected, probably for economic or traditional reasons. This is why the review did not take them into consideration.

  1. “The conclusion can be further improved” The conclusion has been slightly modified.

  1. “What should be the minimum concentrations of the Hg and hydroquinone present in the lightning products?” As reported in the review, in particular in the paragraph Regulations, the main cosmetic regulations, such as the European and American ones, prohibit the use of mercury derivatives as lightening agents in cosmetic products; therefore, there is no minimum concentration of mercury tolerated in skin lighteners. As far as hydroquinone is concerned, it is also forbidden in cosmetics; its use as a skin-lightening active ingredient is confined to the medical field, because it is allowed, at concentrations between 2% and 5%, in galenic and medicinal products to be used under strict medical supervision.

7          What are all the available methods to detect Hg and hydroquinone in commercial samples?”. Several methodologies are available to detect mercury and hydroquinone in skin lighteners, such as spectrophotometric methods, HPLC, UHPLC, voltammetric analysis, as evidenced by recent literature (e.g.: Uddin et al., 2011. Highly sensitive spectrometric method for determination of hydroquinone in skin lightening creams: Application in cosmetics; Harsini et al. 2019. Voltammetric Analysis of Hydroquinone in Skin Whitening Cosmetic Using Ferrocene Modified Carbon Paste Electrode. Rasayan J. Chem., 12, 2296 – 2305; Arshad et al., 2021. Quantitative estimation of the hydroquinone, Mercury, and total plate count in skin-lightening creams. Sustainability, 13, 8786). However, the analytical aspect was not taken into consideration by the Author, because it seemed to be outside the scope of the review.

  • “Please check the grammatical and syntax error”. The text of the article has been checked and several grammatical and syntactic corrections have been made.

Reviewer 2 Report

The article: Diffusion of dangerous skin lightening products as a result of colorism: a review, is an interesting manuscript that presents valuable information about the substances used in cosmetics, thus easily available for a wide range of people but could cause the toxic effect on the human health. The manuscript is well written in a comprehensible way. However, I have some notes which help improve the article.

KEYWORDS

I suggest using the other than in the title words. It makes the article more visible on the websites.

METHODS

I propose the Author incorporate one more paragraph concerning the methods used during the article's writing. This part should describe the sources/internet databases/or the criteria or inclusion criteria on the information presented in the manuscript.

Perhaps Author could better describe arbutin as one of the ingredients of skin lightening cosmetics. Arbutine can be a precursor of hydroquinone. The toxicity with this substance is also possible (the structure of arbutin could also be added), or perhaps the safety is concerned for arbutin? Such information could be added.

The information in part from line 199 is the same as the data from line 216. The Author should perhaps avoid duplication and delete some of it.

The fragment from the 206 – 210 line is rather suitable with the later parts (ex. in line 220, and before the phrase “The above reaction (…) ).

Table 1 does not contain hydroquinone or corticosteroids; why? The Author indicates them as still often used. In addition, the second column in Table 2 is the duplication of information from table 1 – it should be avoided. Perhaps in place table 2, some schema could be more appreciable?

In conclusion, the Author indicates: “The circulation on the world markets of skin lightening cosmetics containing illegal and toxic ingredients represents today a major public health problem…”. I think that the word “major” is some abuse.

I reflect on why Author did not present the Retinoids as the dangerous components of the topical preparations. Perhaps this paragraph should be written. This will expand the importance of the manuscript.

OTHERS

The font type should be unified (the main text and in the Reference).

 Line 207 – the parenthesis is not closed.

Author Response

Author’s Reply to the Review Report (Reviewer 2)

First of all, thank you for taking the time to assess my manuscript

KEYWORDS

 “I suggest using the other than in the title words. It makes the article more visible on the websites.” Thank you for pointing this out. The Author has, accordingly, added some more general keywords to promote the visibility of the article on the websites.

METHODS

“I propose the Author incorporate one more paragraph concerning the methods used during the article's writing. This part should describe the sources/internet databases/or the criteria or inclusion criteria on the information presented in the manuscript.” I appreciate the reviewer’s suggestion; according to it, a sentence (highlighted in red) explaining how the information was found has been added at the end of the paragraph 2.

“Perhaps Author could better describe arbutin as one of the ingredients of skin lightening cosmetics. Arbutin can be a precursor of hydroquinone. The toxicity with this substance is also possible (the structure of arbutin could also be added), or perhaps the safety is concerned for arbutin? Such information could be added.” Thank you for your suggestion. The structure of arbutin and some relevant information on this molecule have been added to the paragraph concerning Hydroquinone. Some clarifications have been added at the end of the same paragraph.

“The information in part from line 199 is the same as the data from line 216. The Author should perhaps avoid duplication and delete some of it.” and “The fragment from the 206 – 210 line is rather suitable with the later parts (ex. in line 220, and before the phrase “The above reaction (…) )”. I agree with this comment, of course; it was an oversight. The information from line 199 has been deleted.

“Table 1 does not contain hydroquinone or corticosteroids; why? The Author indicates them as still often used. In addition, the second column in Table 2 is the duplication of information from table 1 – it should be avoided. Perhaps in place table 2, some schema could be more appreciable?” In the intention of the Author Table 1 should only list the skin-lightening ingredients legally authorized for medicinal and cosmetic purposes. For this reason, mercury derivatives are not included, because they are forbidden; topical corticosteroids were not included because their use as skin lighteners is considered a misuse. Since the caption of table 1 did not clearly express the Author’s intention, it has been modified. As regards Table 2, it refers to misuse of skin-lightening substances and provides information different from Table 1. Therefore the Author respectfully disagrees with the Reviewer’s opinion, and believes that Table 2 should remain as it is.

“In conclusion, the Author indicates: “The circulation on the world markets of skin lightening cosmetics containing illegal and toxic ingredients represents today a major public health problem…”. I think that the word “major” is some abuse.” The word “major” has been replaced with “emerging”.

“I reflect on why Author did not present the Retinoids as the dangerous components of the topical preparations. Perhaps this paragraph should be written. This will expand the importance of the manuscript.” The purpose of the review was to describe the toxic substances more present in skin lighteners popular in countries where the skin bleaching has not only a purely cosmetic meaning, but also has profound cultural and socio-economic implications. In skin-lightening formulations circulating in these countries the banned ingredients traditionally employed are mercury derivatives, hydroquinone, and corticosteroids; no other heavy metals or retinoids have been detected, probably for economic or traditional reasons. This is why the review did not take them into consideration.

OTHERS

 “The font type should be unified (the main text and in the Reference).”  The font type has been unified.

Line 207-the parenthesis is not closed.” The missing parenthesis has been added.